# Amelioration of Cyclosporine A-Induced Acute Nephrotoxicity by *Cordyceps cicadae* Mycelia via Mg^+2^ Reabsorption and the Inhibition of GRP78-IRE1-CHOP Pathway: In Vivo and In Vitro

**DOI:** 10.3390/ijms24010772

**Published:** 2023-01-01

**Authors:** Zong-Han Wu, Chun-Hung Chiu, Chin-Chu Chen, Charng-Cherng Chyau, Chi-Hung Cheng

**Affiliations:** 1Research Institute of Biotechnology, Hungkuang University, Taichung 43302, Taiwan; 2Department of Program in Animal Healthcare, Hungkuang University, Shalu District, Taichung 43302, Taiwan; 3Grape King Biotechnology Center, Longtan District, Taoyuan 325002, Taiwan; 4Department of Nephrology, Catholic Mercy Hospital, Hukou Township 303032, Taiwan

**Keywords:** *Cordyceps cicadae*, liquid cultured mycelium, cyclosporine a, renal injury, endoplasmic reticulum stress, composition analysis

## Abstract

Fruiting bodies of *Cordyceps cicadae* (CC) have been reported to have a therapeutic effect in chronic kidney disease. Due to the rare and expensive resources from natural habitats, artificially cultivated mycelia using submerged liquid cultivation of CC (CCM) have been recently developed as an alternative to scarce sources of CC. However, little is known regarding potential protective effects of CCM against cyclosporine A (CsA)-induced acute nephrotoxicity in vivo and in vitro. In this study, male Sprague-Dawley rats were divided into six groups: control, CCM (40 mg and 400 mg/kg, orally), CsA (10 mg/kg, oral gavage), and CsA + CCM (40 mg and 400 mg/kg, orally). At the end of the study on day 8, all rats were sacrificed, and the blood and kidneys retrieved. CsA-induced acute nephrotoxicity was evident by increased levels of blood urea nitrogen (BUN). Levels of the endoplasmic reticulum (ER) resident chaperone glucose regulated protein 78 (GRP 78) were increased significantly in rats with acute nephrotoxicity. BUN and GRP 78 were significantly ameliorated in synchronous oral groups of CCM (40 or 400 mg/kg) plus CsA. Examination of hematoxylin and eosin stained kidney tissues revealed that the combined treatment of CCM slightly improved vacuolization in renal tubules upon CsA-induced damage. CsA-induced down-regulation of protein expression of magnesium ion channel proteins and transient receptor potential melastatin 6 and 7 were abolished by the combined treatment of CCM. CCM has the potential to protect the kidney against CsA-induced nephrotoxicity by reducing magnesium ion wasting, tubular cell damage, and ER stress demonstrated further by human renal proximal tubular epithelial cell line HK-2. Our results contribute to the in-depth understanding of the role of polysaccharides and nucleobases as the main secondary metabolites of CCM in the defense system of renal functions in CsA-induced acute nephrotoxicity.

## 1. Introduction

Cyclosporine A (CsA) is a calcineurin inhibitor that has been used clinically in organ transplantation since the 1980s [1]. Acute renal dysfunction during CsA therapy is a recognized clinical detriment [2]. The acute nephrotoxicity causes renal vascular resistance and reciprocal renal blood flow decrease, followed by variable impairment of glomerular filtration rate (GFR) [3]. Moreover, the increasing release of endothelin-1, up-regulation of transforming growth factor-beta1 (TGF-β1), and endoplasmic reticulum (ER) stress are also found in the pathogenesis of CsA nephropathy [4]. Some of the important markers of CsA-induced nephrotoxicity are the malfunction in glomerular filtration and magnesium homeostasis and the increase in ER stress in renal tubular cells [3]. Especially, ER stress signaling is one of important pathways involved in the pathogenic role of acute nephrotoxicity [5]. In normal ER function, appropriate protein folding and assembly are expressed in the organelles, whereas pathophysiological insults induce the accumulation of unfolded proteins in ER. Under the stress conditions, especially in acute toxicities, the transcription of unfolded protein response target genes, such as calcium-dependent molecular chaperones that include glucose regulated protein-78 (GRP 78), are activated and overexpressed in response to the physiological conditions [6,7]. 

The impaired magnesium homeostasis by CsA is consequential since magnesium ion (Mg^2+^) is an essential mineral element to maintaining physical health. Magnesium homeostasis depends on the functions of renal excretion and intestinal uptake, which are relative to the proteins expression of the melastatin-related subfamily of transient receptor potential (TRPM) ion channel kinases, including the TRPM 6 and TRPM7 subfamilies [8,9]. TRPM6/7 channels control magnesium entry into distal convoluted tubule epithelial cells through the apical membrane [8]. Thus, expressions of TRPM6/7 proteins influence the regulation of magnesium uptake [10].

Numerous studies have demonstrated that nutraceuticals and diet supplements prevent CsA nephrotoxicity and improve renal function [11,12,13]. *Cordyceps cicadae* (CC) is a parasitic fungus that naturally grows on the cicada nymph or larvae of *Cicada flammate*, *Platypleura kaempferi*, *Cryptotympana pustulata*, and *Patylomia pieli* [14]. CC has been used as a traditional medicine to improve renal function [15,16]. In vitro studies revealed that CC can protect mesangial cells from sclerosis and PC 12 cells from glutamate-induced oxidative damage [17,18]. Moreover, CC extracts provide effective renal protection from both hypertensive renal injury [19] and steroid-induced glaucoma in rats [20]. *C. cicadae* extract ameliorated chronic kidney disease in a subtotal nephrectomy model in rats [16]. In the safety and resource issues, the mycelial products from submerged cultivation of *C. cicadae* (CCM) have been demonstrated to have good safety either in animal or human clinical studies and met the criteria for a potential alternative of the natural product [21,22]. However, little information is available on the specific effects of CCM on cyclosporine A (CsA)-induced nephrotoxicity. The aim of this study was to investigate whether the biochemical changes and renal dysfunctions induced by administration of CsA in rats could be mitigated by concomitant administration of CCM. 

## 2. Results

### 2.1. Proximate Composition and Polysaccharides of CCM

The main dry matter nutrient component of CCM was carbohydrate, constituting 53.48 g/100 g dry weight. The content of water-soluble polysaccharides was 5.15 ± 0.36 g/100 g of CCM, representing a considerable proportion of the carbohydrate compounds. The second most abundant component was crude protein (27.19%). Crude fat and crude ash contributed 7.81% and 4.61%, respectively. 

### 2.2. Nucleobases and Their Derivatives of CCM

For effective extraction of water-soluble secondary metabolites from submerged liquid cultured mycelia of CC, multiple ultrasonic extractions using an aqueous solvent were followed by protein precipitation with acetonitrile. Figure 1 shows the high-performance liquid chromatography (HPLC) chromatogram of the aqueous extract of CCM. Quantitative analyses of nucleobases and their derivatives in the mycelium were performed using the detected peak areas within the HPLC wavelength range of 210–400 nm to assess the content of each compound. The peaks were quantified using an internal method. Compound identification was conducted by mass spectrometry in a positive ionization mode. The obtained molecular ion [M + H]^+^ of each compound was selected for fragmentation experiments with optimum collision energy to produce the product ion to enable structure identification. This identification provided useful information regarding the nucleobases and the composition of their derivatives in the liquid cultured mycelia of CC. A total of 11 compounds, including the nucleobases of cytosine, guanine, and adenine; nucleosides of cytidine, inosine, guanosine, and adenosine; and two nucleoside derivatives, cordycepin, and N^6^-(2-hydroxyethyl) adenosine (HEA), were identified in the CCM extract (Table 1). The guanine content (110.03 μg/g) was the highest, followed by N^6^-(2-hydroxyethyl) adenosine (HEA, 103.15 μg/g) in the extract (Table 1). 

### 2.3. CCM Decreased Proteinuria in CsA-Induced Rats 

The urea and creatinine clearance rates were used to evaluate glomerular function. Oral CsA in rats resulted in the damage of the renal urea clearance rate of blood urea nitrogen (CBUN) compared to the control group (Figure 2, left panel). In CsA + CCM groups, the CsA-induced down-regulation of urea clearance rate was significantly recovered at the low dose of CCM (40 mg/kg). Unexpectedly, the HCC group (400 mg/kg) did not present the same effect as the LCC group. However, the impaired creatinine clearance rate (CrCl) was significantly (*p* < 0.05) improved in the LCC and HCC cotreatment groups compared to the CsA group (Figure 2, right panel). 

### 2.4. CCM Improves Tubular Function in CsA-Administered Rats 

The CsA-mediated damage to renal function was evident in the urea clearance as described above. The fractional excretions of different ions were further used to evaluate the protective effects of CCM on renal tubules. Unusually significant high fractional excretion of magnesium (Mg), calcium (Ca), potassium (K), and phosphorous (P) ions were induced by the administration of CsA (Figure 3A,D). Interestingly, treatment in combination with CCM significantly ameliorated the fractional excretions of Mg, K, and P, but not Ca (Figure 3B), which were impaired by CsA.

### 2.5. CCM Alleviates CsA-Induced Histopathological Changes in Kidney 

Kidney morphology of CsA-induced acute kidney injury (AKI) in rats was further observed by using hematoxylin and eosin (H&E) staining to examine structure abnormalities in CsA-induced kidney damage and the protective effects of CCM on the CsA-induced damages (Figure 4). Kidney sections from the control group (Figure 4) revealed normal histology without damages in the renal corpuscles. However, those treated with CsA (Figure 4) displayed histopathological changes in the renal corpuscles that included shrunken glomeruli (G) and the widening structure of Bowman’s spaces (arrow). As shown in Figure 4 and Figure 5F, the concomitant administration of CCM at doses of 40 and 400 mg/kg with CsA attenuated the structural damage of corpuscles in the absence of the structure damages in the CsA group. Moreover, kidney sections in the CCM alone groups revealed normal structure (Figure 4), indicating that CCM alone does not cause renal damage. 

### 2.6. CCM Up-Regulates TRPM6 and Trpm7protein Expressions Impaired by CsA 

TRPM6 and TRPM7 are involved in the reabsorption of magnesium, which affects the intracellular balance of magnesium. The suppressed expressions of TRPM6 and TRPM7 by CsA would cause renal function injury. Figure 5 shows representative renal immunochemical staining analyses (top panel) and quantitative analyses for TRPM6 and TRPM7 in each group of experimental animals (bottom panel). Quantitative analysis revealed that CsA significantly decreased TRPM6 and TRPM7 (Figure 5, bottom panel). The tissue expressions of TRPM6 and TRPM7 were significantly higher in the CCM-CsA groups for both low and high dosages of CCM compared to that of the CsA group. The histochemistry staining revealed the specific staining of TRPM6 and TRPM7 in the border of distal renal tubule cells (Figure 5, top panel). In the CsA group, both TRPM6 and TRPM7 exhibited much lower expression than those of control and CCM-CsA groups. The findings suggest that CCM ameliorated TRPM6 and TRPM7 expressions that had been impaired by CsA.

### 2.7. CCM Reduces Expression of the ER Stress Marker In Vivo 

The overexpression of GRP 78 or immunoglobulin heavy chain binding protein (BiP) is directly related to ER stress activity. When applied, even at the low dosage of 40 mg/kg, CCM ameliorated the effects of CsA. Therefore, we investigated the association of GRP 78 protein expression in the CsA- and CCM-treated groups. Western blot results revealed that CsA significantly induced the expression of GRP 78 (Figure 6). The combined treatment with CsA and CCM significantly (*p* < 0.05) reversed the effects of CsA. A slight but significant (*p* < 0.05) increase occurred with treatment solely with CCM compared to control group (in the % of 52.3 ± 4.6 vs. 28.2 ± 9.5). 

### 2.8. Effect of CCM and CsA on Cell Viability in HK-2 Cells

To assess whether CCM was toxic to HK-2 cells, the effects of different concentrations of CCM (0, 100, 300, and 600 µg/mL) on cell viability were examined. The cell viability preserved up to 600 μg/mL CCM without any significant toxicity for 24 h (Figure 7A). Subsequently, HK-2 cells were treated with doses of 100 and 300 µg/mL CCM for 24 h posterior to CsA treatment for 2 h. Furthermore, these assays revealed that treatment with CsA (10 and 20 nM) led to a significant reduction after 48 and 72 hr incubation in a time and dose-responsive manner (Figure 7B). The choice of CsA concentration at 10 nM was then determined for the following experiments which meet the criteria in a comparable cell viability (80% viability) and potential capability for inducing the ER stress. 

### 2.9. CCM Reduces Expression of the ER Stress Marker In Vitro

To further investigate whether ER stress is associated with CsA intervention, the activation of ER stress markers, including CHOP, ATF6, IRE1α, and PERK, were investigated in the human renal cell line HK-2. The m-RNA expression levels demonstrated the levels of the above ER stress markers were significantly increased in the CsA-treated cells, while CCM treatment without any significant effect and partly increasing inhibition was detected in the treatments (Figure 8A,B,D). However, the findings of the current study showed that either CCM or CsA lost its effect on ATF6 (Figure 8C) on one of three main signaling systems of ER stress markers, indicating that the HK-2 cells cotreatment with CCM and CsA at the indicated time was specifically effective in the IRE1- and PERK-associated signaling systems. Thus, the exact role of CCM on the effect of ATF6 expression in the study is not completely established. Further investigation would be warranted to examine and explain the phenomenon. 

## 3. Discussion 

The results of this study showed the potential value of CCM produced by liquid fermentation in renal protection against CsA-induced AKI in rats and human renal cells. In previous studies, a 90-day subchronic toxicity study of CCM revealed no statistical differences in body weight gain, relative organ weight, hematology, serum chemistry, and urinalysis [21]. A safety assessment of CCM from 49 participants reported no side effects [22]. The clinical study results extend the potential value of CCM from food safety to the development of alternative and complimentary medicines. Concerning efficacy, the biological activities are correlated with the complicated composition of CCM. In this study, we have simultaneously investigated the chemical components of CCM. This comprehensive investigation included the proximate (general) composition, polysaccharide content, and composition of the main nucleobases and their derivatives (Table 1). As expected, products of submerged liquid cultures of CC presented the highest amount of carbohydrate in proximate composition. Relative findings have been observed in *Cordyceps* species [24] and mycelia of *C. militaris* [25] and *C. sobolifera* [9]. Concerning the carbohydrate compositions, the high percentage of CCM water-soluble polysaccharides (5.15% on a dry weight basis) may play an important role in physiological functions, including the efficacy in enhancing the immune system [26]. In the identification of bioactive small molecules in inhibiting detrimental factors, induced ER stress has attracted attention of researchers and scientists. We found that the important bioactive compound HEA (Table 1) was identified in the CCM similar to that of fruiting bodies of CC [17]. In addition, cordycepin was also found in the populations of CC, including the different species of *C. militaris* [14]. HEA has been indicated as one of the important bioactive compounds in CCM in ameliorating nonsteroidal anti-inflammatory drug-stimulated ER stress in human renal cells [27]. It also attenuates the lipopolysaccharide-induced proinflammatory responses by suppressing the Toll-like receptor 4-mediated nuclear factor-kappa B (NF-κB) signaling pathway [28]. More recently, HEA was reported to have ameliorative capabilities against interstitial fibrosis in mice induced by unilateral ureteral obstruction [29].

According to the previous report [30], CsA treatment is associated with stimulation of oxygen radical formation in liver and kidney. Moreover, the results of the treatment of HK-2 cells with CsA significantly increased ROS and malondialdehyde levels and decreased the activities of SOD, GSH-Px, and CAT compared with the control group, while the antioxidant curcumin abolished CsA-induced nephrotoxicity in rats and HK-2 cells [31]. We suggested that the inhibition of ER stress might be related to the antioxidant of cordycepin, a compound isolated from CCM. The isolated compound has been shown to markedly reduce cellular malondialdehyde (MDA) content and intracellular reactive oxygen species (ROS) level effects in neuron cells [32].

Although CsA is a well-known immunosuppressive agent used in organ transplantation, some adverse effects have been revealed in the clinical studies [2,3,4]. In this study, the dose (10 mg/kg) and administration period (7 days) of CsA were selected based on the proven effectiveness in inducing nephrotoxicity in a previous similar study [9]. CsA nephrotoxicity causes disorders in renal function in the fractional excretion (FE) of uric acid and minerals [1,3]. Thus, FE of minerals have long been considered as an excellent research tool to investigate tubular physiology in the diagnoses of kidney disorders [33,34]. Hypomagnesemia and increased magnesium excretion were observed after long-term use of CsA [35]. In an investigation of patients at different stages of chronic kidney disease, fractional excretion (FE) of electrolytes, including potassium, calcium, phosphorus, and magnesium ions, tended to increase along with the decline of renal function [36]. Therapeutic effects of *C. cicadae* have been shown with significant clinical effects on chronic renal failure [37]. Furthermore, it has been demonstrated that the bioactive compounds cordycepin, adenosine, and inosine of *C. cicadae* extracts contributed to the effects in maintaining the serum electrolytes homeostasis from drug-induced renal toxicity [38]. Our results showed that CsA impaired tubule function by increasing the FE of magnesium, and CCM significantly reduced the FE of magnesium (Figure 3A). Similar results were previously observed [9]. This study provides explanations to the administration of CCM in inhibiting CsA-induced nephropathy through the inhibition of magnesium wasting and ER stress detrimental mechanisms in vivo and in vitro.

The control of Mg^2+^ transport across cell membranes is related to the expressions of TRPM6 and TRPM7 in the distal tubule membrane [39]. Sawada and coworkers indicated that the suppression of magnesium secretion proteins TRPM6 and TRPM7 in the distal renal tubule induced by CsA treatment may affect the bioavailability of magnesium ions [39]. We observed that the CsA-mediated suppressions of TRPM6 and TRPM7 were recovered partly by the CCM treatment (Figure 5). This might be the potential mechanism by which CCM restores FE Mg impaired by CsA.

CsA nephrotoxicity characteristically features tubular vacuolization [40], with an ER stress origin in collaboration with the increasing expression of GRP78 and C/-EBP homologous protein (CHOP) [41,42]. The induction of GRP78 has been reported as a central regulator for ER stress due to its role as a major ER chaperone with its ability to control the activation of transmembrane ER stress sensors (IRE1, PERK, and ATF6) through a binding–release mechanism [43]. Moreover, the invention of harmful stimuli induced the ER transmembrane proteins, PERK, ATF6, and IRE1, to separate from the chaperone GRP78 and stimulate downstream transcriptional target CHOP to be activated [36]. In the study, CsA induced tubular vacuolization and irregular cells with the destruction of the brush border and overexpression of GRP 78 (Figure 6). The CCM cotreated group displayed milder histopathological changes in a more or less similar manner to the control group (Figure 4). Thus, the inhibition of crosstalk between GRP78 and TRPM6 and seven channels might be the potential targets for the treatment of AKI. This study may have a translational impact and contributes to the basic knowledge on nutraceutical preparation and application of CCM in the future.

## 4. Materials and Methods

### 4.1. Chemicals

Acetonitrile, methanol, n-butanol, and ethyl acetate were of the HPLC grade, and guanine, adenine, inosine, guanosine, and 8-bromoadenosine-3′,5′-cyclic monophosphate standards applied in the analysis of nucleosides; nucleotides were purchased from Sigma-Aldrich (St. Louis, MO, USA). 

### 4.2. Cultivation and Preparation of C. cicadae Mycelium (CCM)

The cultivation of *C. cicadae* was conducted at the Grape King Biotech Research Institute (LongTan, Taoyuan City, Taiwan), as previously described [21,22]. In brief, *C. cicadae* strain (MU30106) obtained from the Food Industry Research and Development Institute (FIRDI, Hsinchu, Taiwan) was grown on potato dextrose agar and was incubated at 25 °C for 5 days. Then, a 5 mm × 5 mm piece of *C. cicadae* agar culture was inoculated into a 2 L Hinton flask containing 1.0 L of potato dextrose broth and was incubated at 25 °C in a 120 rpm rotary shaker for 5 days. The starter was then scaled up to 120 L (2% glucose, 1% yeast extract, 1% soybean powder; pH 6.0) in a 200 L fermenter (BioTop Co., Taichung, Taiwan) and agitated at 60 rpm with an aeration rate of 0.5 vvm at 25 °C for 3 days. Finally, the submerged mycelial culture was heated at 100 °C for 1 h, freeze-dried, powdered, and stored at −30 °C for further study. 

### 4.3. Proximate Composition Analysis and Polysaccharide Extraction 

The proximate composition analysis on macronutrients of freeze-dried CCM powder was performed on crude protein, total crude lipids, ash, fiber, and moisture contents based on the standard methods of the Association of Official Analytical Chemists methods 984.13, 43.275, 968.08, 991.43, and 950.46.B, respectively [44]. 

Water-soluble polysaccharides were extracted according to previous report [9]. Freeze-dried and defatted mycelia powder (10 g) was extracted with reflux three times with 200 mL volumes of double-distilled water at 90 °C for 2 h. The obtained extracts after evaporation at 40 °C in a rotary evaporator were added into ethanol (95%) in a 1:2 ratio (*v/v*) to precipitate the water = soluble polysaccharides, which were collected and further purified in addition to a 3-fold volume of ethanol (95%) and then collected and lyophilized to obtain the polysaccharides (CCP). Contents of polysaccharide and protein were determined by the phenol-sulfuric acid method [45] and the Lowry method [46], respectively. 

### 4.4. HPLC and Liquid Chromatography-Tandem Mass Spectrometry (LC-MS/MS) Analyses of Aqueous Extracts of CCM

The powdered mycelia (1.0 g) were dissolved in 10 mL distilled water containing the internal standard 8-bromoadenosine-3′,5′-cyclic monophosphate (5.0 μg; Sigma–Aldrich, St. Louis, MO, USA) and was subjected to sonication using a Delta Sonicator DC200H (LMI Co. Ltd., Taipei, Taiwan) at 40 °C for 60 min. The extraction was performed in triplicate. Obtained extracts (30.0 mL) concentrated to 10 mL using a rotary vacuum evaporator at 40 °C. After mixing the aqueous extracts with acetonitrile (10 mL), the mixture was placed in a refrigerator (4 °C) overnight. The supernatant was collected after the centrifugation at 10,000× *g* for 15 min and was transferred to an evaporating flask for further evaporation of solvent to yield 2 mL concentrates. The concentrates were analyzed by LC-MS on a series 1260 Infinity HPLC system (Agilent Technologies, Santa Clara, CA, USA) equipped with a model G1379B degasser, model G1312B binary gradient pump, model G1329B autosampler, model G1316A column oven (at 35 °C), and model G1315D photodiode array detection (PDA) system. Chromatographic separation was conducted on an Acquity HSS T3 C18 analysis column (2.1 × 150 mm; 1.8 μm particle size, Waters Corp., Milford, MA, USA) and a precolumn (SceurityGuard C18 (ODS) 4 mm × 3.0 mm ID, Phenomenex Inc., Torrance, CA, USA). The mobile phase consisted of 50 mM ammonium formate in water (A) and acetonitrile (containing 0.1% formic acid) (B), with a gradient elution as follows: 0–3 min, 2–5% B; 3–12 min, 5–30% B; 12–30 min, 30–70% B; 30–35 min, 70–2% B. The flow rate was maintained at 0.2 mL/min. The photodiode array detector (DAD) was set at 260, 280, and 320 nm in the scan range of 210–400 nm. The separated components were then introduced into the Agilent 6420 quadruple MS system equipped with an electrospray ionization (ESI) interface and operated in positive ionization mode in a potential of 3500 V. High purity of nitrogen (99.999%) was used as the drying gas (350 °C) at a flow rate of 8 L/min, and the nebulizer gas was set at a pressure of 35 psi. The Agilent MassHunter Workstation B.01.04 Software was used for the data collection and analysis. Compound identification was performed according to previous reports [23,47] using selected reaction monitoring, as shown in Table 1. 

### 4.5. Animal Studies

In our previous study [9], nephrotoxicity was observed in male Sprague-Dawley (SD) rats exposed to CsA. Based on the study, we applied the procedures to explore the hypothesis that CCM extracts might ameliorate the symptoms of CsA-induced nephrotoxicity. Thirty-six male SD rats, 8 weeks of age, were obtained from BioLASCO (A Charles River Licensee Corp., Yi-Lan, Taiwan). The experiment complied with the regulation controlled by the Animal Research Committee of Hungkuang University (HKU). All experimental protocols were approved by the Ethical Committee of HKU (No. HK-P-102025). The rats were housed in a room with controlled temperature (23 ± 2 °C) and relative humidity (55 ± 5%), with 12 h light–dark cycles. Rats were fed standard laboratory chow and given reverse osmosis water ad libitum. After a one-week acclimatization, rats were randomly divided into six groups (n = 6 each) and assigned to the control, CsA, LCC alone, HCC alone, LCC + CsA, and HCC + CsA groups, respectively. For nasogastric tube feeding, all of the samples were dissolved in 0.9% saline. The LCC and HCC groups received CCM at low and high doses of 40 and 400 mg/kg body weight, respectively, for 7 days. The CsA group also received a daily oral gavage of CsA (10 mg/kg; Novartis Pharma, Basel, Switzerland) for 7 days. In the LCC + CsA and HCC + CsA groups, animals were treated with CsA (10 mg/kg). Six hours later, LCC (40 mg/kg) and HCC (400 mg/kg) were orally administered. All treatments were administered daily for 7 days. Animals in the control group received 0.9% normal saline. Before sacrifice, rats were housed individually in metabolic cages to collect 24 h urine. After blood sampling, half of the kidney was dissected and fixed in 4% buffered paraformaldehyde at room temperature, dehydrated with alcohol, and embedded in paraffin. The other half of the kidney was dissected and frozen at −80 °C until analysis. 

### 4.6. Biochemical Analysis

Serum and urine biochemical analyses, including BUN, calcium, phosphate, magnesium, and sodium, were analyzed as described previously [9]. FE parameters included magnesium ion, (FE Mg), sodium ion (FE Na), calcium ion (FE Ca), and uric acid (FE uric acid). Levels of creatinine clearance rate and urea nitrogen renal clearance of urine and serum were also determined as described previously [9]. 

### 4.7. Histological Analysis

Renal specimens from all animals were embedded in paraffin and cut into 2 µm-thick sections on a rotary microtome (Leica Microsystems, Herlev, Denmark), then stained with hematoxylin (1%, *v/v*) and eosin (0.5% *v/v*). Kidney damage was then assessed by light microscopy (Olympus, Tokyo, Japan). 

### 4.8. Immunohistochemistry Staining of TRPM6 and TRPM7

Renal tissue was blocked in paraffin, and sliced array specimens were created at thicknesses of 3 μm. The formalin-fixed specimens were washed in Tris-buffered saline (TBS) containing 0.1% Tween-20 and rehydrated through graded ethanol to phosphate buffered saline (PBS, pH 7.2). Sections were incubated with primary antibodies to TRPM6 (LifeSpan BioSciences Inc., Seattle, CA, USA) and TRPM7 (GeneTex Inc. Irvine, CA, USA) (both diluted 1:200 in TBS) for 15 min at room temperature. After washing in TBS, the section was treated with a horseradish peroxidase-polymer-conjugated secondary antibody (1:200 in TBS; Dako, Carpinteria, CA, USA). Color development was performed with diaminobenzidine hydrochloride (DAB) as the chromogen [47] and was visualized by light microscopy. To quantify TRPM6 and TRPM7 in the kidney, a total of 10 consecutive high-power fields (×200) were examined by microscopy and photographed. A grid containing 100 (10 × 10) sampling points was randomly chosen from each and superimposed on each photograph. The image morphometry of special staining (TRPM6 and TRPM7) was determined by using ImageJ software. 

### 4.9. Western Blot Analysis of GRP 78 in Rat Kidney 

A previously described experimental procedure [9] was used. In brief, 70 µg of protein sample from renal tissues was resolved by 9% sodium dodecyl sulfate–polyacrylamide gel for electrophoresis. After the electrophoresis, the proteins were transferred to a polyvinylidene fluoride membrane. The membrane was blocked with TBS buffer (20 mM Tris–HCl, 150 mM NaCl, pH 7.4) containing 5% nonfat milk at room temperature for 60 min and then incubated with rabbit GRP 78 antibody (1:500 in PBS, ab21685, Abcam, Cambridge, UK). The membrane was washed three times with PBST, and then antirabbit immunoglobulin-labeled horseradish peroxidase-conjugated polyclonal antibody was added and detected. Fujifilm LAS 3000 was used for a quantitative analysis of the results of the immunoblot. The images were analyzed under Multi Gauge V3.0 (Life Science Systems, Fujifilm Global). 

### 4.10. In Vitro Study of m-RNA Expression

To investigate whether the CCM could reduce the CsA-induced ER stress markers, the human renal proximal tubular epithelial cell line HK-2 was further applied to the studies. The cell cultivation and cell viability assayed by the trypan blue exclusion method were described in the previous report [27]. For gene expression studies, The HK–2 cells at a density of 5 × 10^5^ cells/mL were seeded onto a 6 cm dish and incubated for 24 h. HK-2 cells were preincubated with CsA at 10 nM for 2 h, followed by treatment with CCM extracts at 100 and 300 μg/mL, respectively, for another 24 h. N-acetylcysteine (NAC, 1 mM) was used as a positive control. Total RNA was isolated from cells using the TRIzol^®^ Reagent (Thermo Fisher Scientific, Waltham, MA, USA) according to the manufacturer’s protocol. In reverse transcription of RNA to cDNA, total RNA (1.5 µg) was reverse-transcribed using Takara PrimeScript^TM^ RT-PCR Reagent Kit (Takara Bio, Mountain View, CA, USA). An equal amount of cDNA was used for the subsequent qPCR performed with the SYBR^®^ FAST (KAPA biosystems). Amplification was performed in a StepOnePlus™ Real-Time PCR System (Applied Biosystems, Foster City, CA, USA). The DNA fragments were amplified for 40 cycles (enzyme activation: 20 s at 95 °C hold; denaturation: 3 s at 95 °C; annealing: 40 s at 60 °C). The expression of β-actin was determined as the internal control. Relative expression level was calculated using the 2^−∆∆Ct^ method. Appendix A Appendix A depicts the sequence of primers used in the study.

### 4.11. Statistical Analyses

Statistical analyses were performed using SPSS version 15.0 software (SPSS, Inc., Chicago IL, USA). Statistical comparisons between study groups were analyzed by one-way ANOVA, followed by the Fisher’s protected least significant difference (LSD) tests. *p*-values < 0.05 were considered statistically significant. 

## 5. Conclusions 

The present study reveals that the CCM extracts rich in nucleosides and polysaccharides can ameliorate CsA-induced renal toxicity by balancing the Mg^+2^ metabolisms through the enhancement expressions of TRPM6 and TRPM7. We found simultaneous treatment of CCM is beneficial up to a certain extent to alleviate the ER stress-induced damages through the regulation of GRP 78 protein expression to rescue the damaged tissues. Further studies are warranted to gain a better understanding of the underlying mechanisms and regulatory pathways of CCM in renal diseases. 

## Figures and Tables

**Figure 1 ijms-24-00772-f001:**
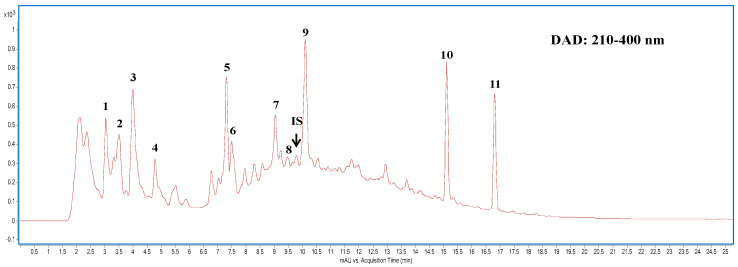
Highperformance liquid chromatography (HPLC) chromatogram of *Cordyceps cicadae* mycelia extract detected at full-range wavelength (210–400 nm) absorption by photodiode array detector. Peak numbers are described in Table 1.

**Figure 2 ijms-24-00772-f002:**
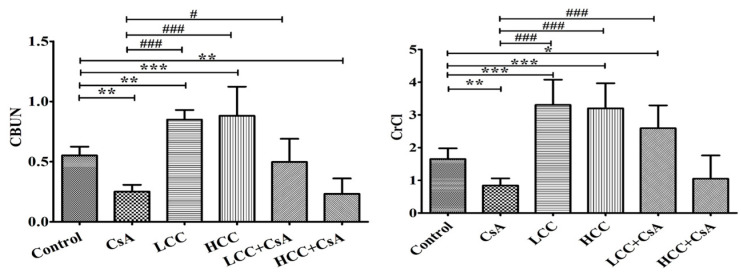
Effects of cyclosporine A (CsA) and CCM treatment on serum creatinine and urea levels. Control, CsA (10 mg/kg), LCC (40 mg/kg), HCC (400 mg/kg), LCC (40 mg/kg) + CsA (10 mg/kg), HCC (400 mg/kg) + CsA (10 mg/kg). Values are expressed as the mean ± SEM (n = 6). One-way analysis of variance (ANOVA) is followed by the Fisher’s protected least significant difference (LSD) tests. * *p* < 0.05, ** *p* < 0.01, and *** *p* < 0.001 vs. control. ^#^
*p* < 0.05 and ^###^
*p* < 0.001 vs. CsA group.

**Figure 3 ijms-24-00772-f003:**
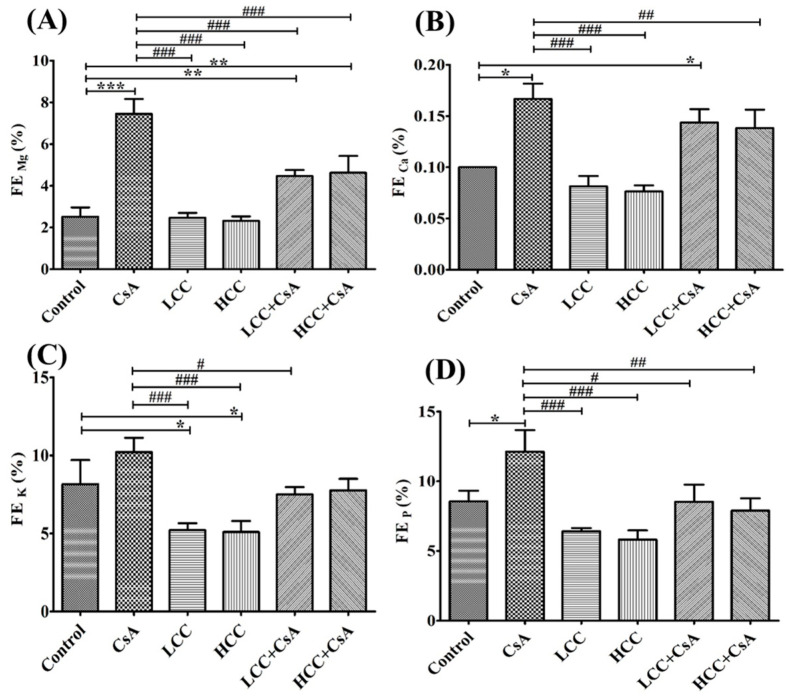
Effects of cyclosporine A (CsA) and CCM treatment on fractional excretion (FE) of Mg (**A**), Ca (**B**), K (**C**), and P (**D**) ions in serum. Control, CsA (10 mg/kg), LCC (40 mg/kg), HCC (400 mg/kg), LCC (40 mg/kg) + CsA (10 mg/kg), and HCC (400 mg/kg) + CsA (10 mg/kg). Values are expressed as the mean ± SEM (n = 6). One-way ANOVA is followed by the Fisher’s LSD tests. * *p* < 0.05, ** *p* < 0.01, and *** *p* < 0.001 vs. control. ^#^
*p* < 0.05, ^##^
*p* < 0.01 and ^###^
*p* < 0.001 vs. CsA group.

**Figure 4 ijms-24-00772-f004:**
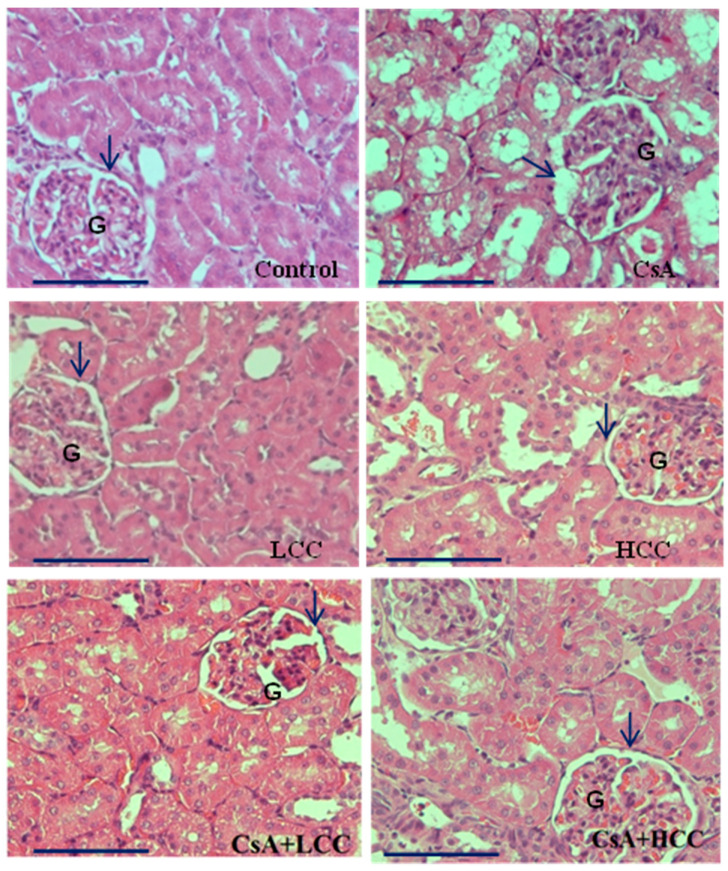
Representative photomicrographs of H&E stained sections in the renal cortex. Control group shows normal architecture of glomerulus (G) and the urinary space (arrow). CsA group shows disturbed architecture of renal cortex in a shrinkage pattern of the glomerulus (G) and widening of the urinary space (arrow). LCC (40 mg/kg) and HCC (400 mg/kg) groups present similar architecture of the renal cortex to that of control group with intact glomerulus (G) and urinary space (arrow). The CsA + LCC group displays partly preserved architecture of the glomerulus (G) and less widening of urinary space (arrow) than the CsA group. The CsA + HCC group displays further amelioration of the glomerulus (G) without shrinkage pattern and an intact urinary space (arrow). Magnification, 400× *g* (bar size 20 μm).

**Figure 5 ijms-24-00772-f005:**
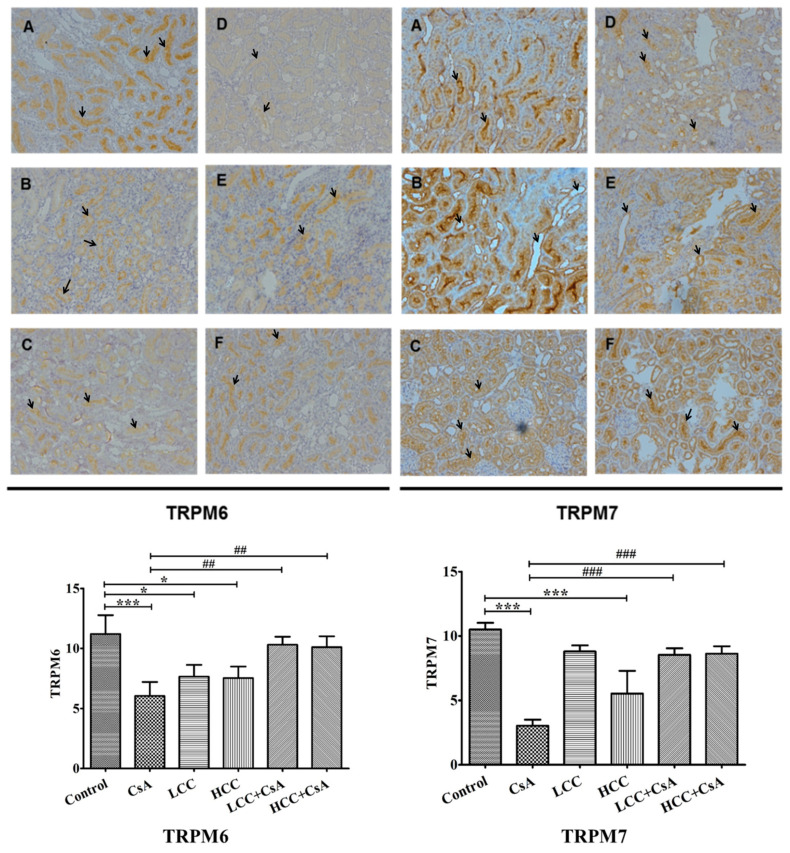
Representative immunohistochemistry (top panel) and quantitative analysis (bottom panel) of the levels of transient receptor potential melastatin 6 (TRPM6) and TRPM7 in rats treated with CCM and CsA. Arrows indicate the cells with the differences of TRPM6 and TRPM7 expression among groups. Values are expressed as the mean ± SEM (n = 6). Control (**A**), CsA (**B**), LLC (40 mg/kg) (**C**), HCC (400 mg/kg) (**D**), LCC (40 mg/kg) + CsA (**E**), and HCC (400 mg/kg) + CsA (**F**). One-way ANOVA is followed by the Fisher’s LSD tests. * *p* < 0.05 and *** *p* < 0.001 vs. control. ^##^
*p* < 0.01 and ^###^
*p* < 0.001 vs. CsA group.

**Figure 6 ijms-24-00772-f006:**
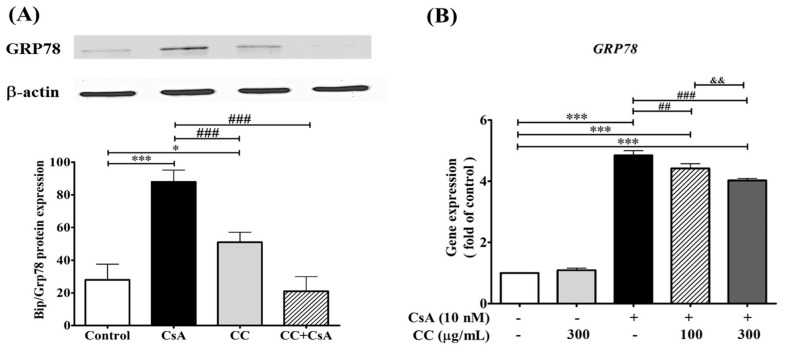
*Cordyceps cicadae* mycelia extract (CCM) protects cell from cyclosprorine A (CsA)-induced expression of glucose regulated protein 78 (GRP 78) in rat renal cells (**A**) and human renal cell line HK-2 cells (**B**). (**A**): The protein expression of GRP 78 increased during CsA challenge, compared to the control, and was counteracted by CCM cotreatment (CC + CsA) in rat. Each column represents the mean ± SEM (n = 6). Different letters indicate a significant difference (*p* < 0.05). (**B**): Human renal cell line HK-2 cells were treated with CsA (10 nM) for 2 h and then incubated with 100 and 300 μg/mL of CCM for 24 h. The m-RNA expression of GRP 78 of human renal cell line HK-2 cells was inhibited by CCM treatment in a dose-responsive manner. Data are presented as mean values ± S.D of three experiments. * *p* < 0.05 and *** *p* < 0.001 vs. control. ^###^
*p* < 0.001 and ^##^
*p* < 0.01 vs. CsA group. ^&&^
*p* < 0.01.

**Figure 7 ijms-24-00772-f007:**
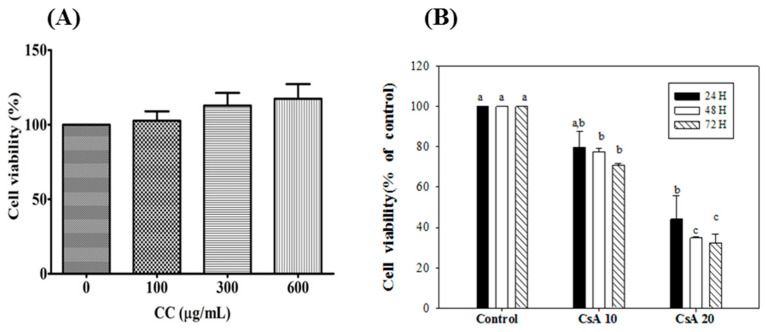
Effects of CCM (**A**) and Csa (**B**) on viability of HK-2 cells. Each value represents the mean ± SEM (n = 3). Different letters indicate significant differences between groups (*p* < 0.05).

**Figure 8 ijms-24-00772-f008:**
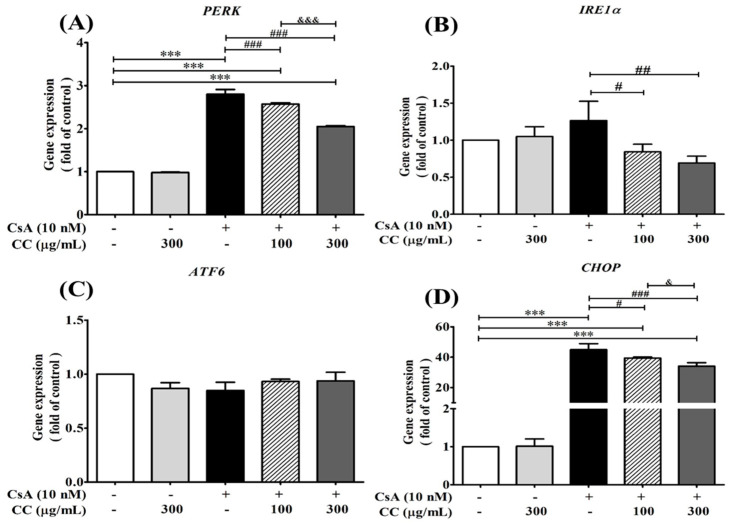
*Cordyceps cicadae* mycelia extract (CCM) protects cell from cyclosprorine A (CsA)-induced cell proliferations and m-RNA expressions of ER stress relative mediators PERK (**A**), IRE1α (**B**), ATF6 (**C**), and CHOP (**D**) in human renal proximal tubular epithelial cell line HK-2. HK-2 cells were treated with CsA (10 nM) alone or together with CCM (100 and 300 μg/mL) at 70–80% confluence for 24 hr. Total RNA was extracted using the TRIzol^®^ reagent, and mRNA levels were measured by quantitative RT-PCR and normalized by β-actin mRNA levels. Data are presented as means ± SD (n = 3). *** *p* < 0.001 vs. control; ^###^
*p* < 0.001, ^##^
*p* < 0.01, and ^#^
*p* < 0.05 vs. the CsA group; ^&&&^
*p* < 0.001 and ^&^
*p* < 0.05 vs. the CC group.

**Table 1 ijms-24-00772-t001:** HPLC-DAD-ESI-MRM analysis on the nucleobases their derivatives compositions of aqueous extract of *Cordyceps cicadae* mycelia.

PeakNo. ^a^	RT (min)	Assigned Identity	Precursor Ion, *m/z*	Product Ion, *m/z*	Fragmentor (V)	Collision Energy (V)	Content (μg/g)
1	3.03	Cytosine ^c^	112	95	160	20	54.65
2	3.50	Cytidine ^c^	244	112	110	10	33.43
3	3.99	Guanine ^b^	152	135	160	20	110.03
4	4.78	Adenine ^b^	136	119	160	20	20.13
5	7.31	Inosine ^c^	269	137	115	10	60.14
6	7.50	Guanosine ^b^	284	152	140	10	15.12
7	9.05	Adenosine ^b^	268	136	135	20	28.04
8	9.47	Cordycepin ^c^	252	136	135	20	6.17
IS	9.79	IS ^e^	409	215	180	20	5.00
9	10.10	HEA ^b^	312	180	135	20	103.15
10	15.11	M.W. 430 ^d^	431	255	125	15	90.16
11	16.82	M.W. 446 ^d^	447	271	125	15	78.01

^(a)^ Peak numbers refer to those in Figure 1. ^(b)^ Compounds were identified by comparison with authentic standards. ^(c)^ Compounds were tentatively identified based on the selected reaction monitoring spectra published in the literature. [23]. ^(d)^ Unidentified compounds in limited MS and MS-MS identification. ^(e)^ Internal standard: 8-bromoadenosine-3′,5′-cyclic monophosphate.

## Data Availability

The data presented in this study are available from the corresponding author upon request.

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
