# Peer review of "Amelioration of Cyclosporine A-Induced Acute Nephrotoxicity by Cordyceps cicadae Mycelia via Mg+2 Reabsorption and the Inhibition of GRP78-IRE1-CHOP Pathway: In Vivo and In Vitro"

_ijms, 2023, doi:10.3390/ijms24010772_

Round 1
Reviewer 1 Report
The manuscript used liquid fermentation Cordyceps militaris mycelia and obtained CCM. The authors designed a series of experiments to verify the effect of CCM on kidney dysfunction.
Several problems need to be added and considered by the authors:
1. The figures in the manuscript are not uniform, includes color, style, significance markers and coordinate names.
2. Control group and CsA group were not check significance marker in any figure.
3. Please mark the changes in Fig 5 (IHC) which described in the manuscript.
4. Protein 78 (GRP 78) or m-RNA expression of GRP 78 in vivo are related to ER stress activity, but this one protein and one gene may not reflect the changes of ER activity, authors should think more genes or protein to prove the function.
5. Are there the same genes that reflect ER activity in rats and HK-2 ? The genes that respond to ER stress should have overlap in rats and HK-2.
Author Response
Please read the attached file

Reviewer 2 Report
This manuscript describes a well-designed study in which a possible nephroprotective effect of CC is postulated in the face of acute aggression caused by the oral administration of cyclosporine A (CsA), concluding that a reduction in renal damage is produced, fundamentally by a CC intervention in Renal Mg++ homeostasis.
In my opinion, the study is well developed, although it could be improved in some aspects:
- Some studies should be included in which other compounds with nephroprotective capacity act on Mg++ homeostasis to increase the importance of the results obtained.
- The last sentence of the introduction would be more appropriate to place it at the end of the discussion. This section should conclude with clear and simple objectives.
- In addition to a qualitative analysis of the histological images (hematoxylin-eosin staining), a morphometric study should be carried out to give more accuracy to the results obtained.
- In the discussion, other mechanisms of renal damage caused by CsA should be mentioned, such as an increase in oxidative stress in renal tissue or the induction of hepatorenal syndrome.
- Finally, the excipient of the CsA should be indicated since they are usually lipid in nature. Castor oil or castor oil, both used in different CsA preparations, affect inflammatory parameters and oxidative stress, which could interfere with the interpretation of results. Do you have information about a group of rats treated with the excipient? I think it is an important aspect.
Author Response
Please read the attached file.

Round 2
Reviewer 1 Report
感谢作者对我问题的回答。祝愿大家今后在这个研究方向上取得更多的科学成就。
Reviewer 2 Report
.